# COVID-19, Vaccines, and Thrombotic Events: A Narrative Review

**DOI:** 10.3390/jcm11040948

**Published:** 2022-02-11

**Authors:** Maurizio G. Abrignani, Adriano Murrone, Leonardo De Luca, Loris Roncon, Andrea Di Lenarda, Serafina Valente, Pasquale Caldarola, Carmine Riccio, Fabrizio Oliva, Michele M. Gulizia, Domenico Gabrielli, Furio Colivicchi

**Affiliations:** 1Cardiology Department, P.O. Sant’Antonio Abate, ASP Trapani, 91016 Erice, Italy; 2Cardiology-UTIC, Hospitals of Città di Castello and Gubbio-Gualdo Tadino, AUSL Umbria 1, 06100 Perugia, Italy; adriano.murrone@gmail.com; 3Cardiology, Cardio-Thoraco-Vascular Department, San Camillo Forlanini Hospital, 00100 Rome, Italy; leo.deluca@libero.it (L.D.L.); dgabrielli@scamilloforlanini.rm.it (D.G.); 4Cardiology Department, Santa Maria della Misericordia Hospital, 45100 Rovigo, Italy; roncon.loris@gmail.com; 5Cardiovascular and Sports Medicine Department, Azienda Sanitaria Universitaria Giuliano Isontina-ASUGI, 34100 Trieste, Italy; dilenar@units.it; 6Clinical Surgical Cardiology (UTIC), A.O.U. Senese, Santa Maria alle Scotte Hospital, 53100 Siena, Italy; seravalente@gmail.com; 7Cardiology-UTIC, San Paolo Hospital, 70100 Bari, Italy; pascald1506@gmail.com; 8Follow-Up of the Post-Acute Patient Unit, Cardio-Vascular Department, A.O.R.N. Sant’Anna and San Sebastiano, 81000 Caserta, Italy; carmine.riccio8@icloud.com; 9Cardiology 1-Hemodynamics, Cardiological Intensive Care Unit, Cardiothoracovascular Department “A. De Gasperis”, ASST Grande Ospedale Metropolitano Niguarda, 20100 Milan, Italy; fabrizio.oliva@ospedaleniguarda.it; 10Cardiology Department, Garibaldi-Nesima Hospital, Company of National Importance and High Specialization “Garibaldi”, 95100 Catania, Italy; michele.gulizia60@gmail.com; 11Heart Care Foundation, 50121 Florence, Italy; 12Clinical and Rehabilitation Cardiology Department, Presidio Ospedaliero San Filippo Neri—ASL Roma 1, 00100 Rome, Italy; furio.colivicchi@gmail.com

**Keywords:** COVID-19, vaccines, thrombosis, coronavirus, SARS-CoV-2, cerebral venous thrombosis, COVID-19 vaccination, thrombotic thrombocytopenia syndrome, vaccine-induced thrombotic thrombocytopenia, platelet factor-4

## Abstract

The coronavirus disease 2019 (COVID-19), a deadly pandemic that has affected millions of people worldwide, is associated with cardiovascular complications, including venous and arterial thromboembolic events. Viral spike proteins, in fact, may promote the release of prothrombotic and inflammatory mediators. Vaccines, coding for the spike protein, are the primary means for preventing COVID-19. However, some unexpected thrombotic events at unusual sites, most frequently located in the cerebral venous sinus but also splanchnic, with associated thrombocytopenia, have emerged in subjects who received adenovirus-based vaccines, especially in fertile women. This clinical entity was soon recognized as a new syndrome, named vaccine-induced immune thrombotic thrombocytopenia, probably caused by cross-reacting anti-platelet factor-4 antibodies activating platelets. For this reason, the regulatory agencies of various countries restricted the use of adenovirus-based vaccines to some age groups. The prevailing opinion of most experts, however, is that the risk of developing COVID-19, including thrombotic complications, clearly outweighs this potential risk. This point-of-view aims at providing a narrative review of epidemiological issues, clinical data, and pathogenetic hypotheses of thrombosis linked to both COVID-19 and its vaccines, helping medical practitioners to offer up-to-date and evidence-based counseling to their often-alarmed patients with acute or chronic cardiovascular thrombotic events.

## 1. Introduction

Since the beginning of the SARS-CoV-2 pandemic, and the consequent coronavirus disease 2019 (COVID-19), up to date (6 February 2022), more than 394 million cases and over 5.7 million deaths have been documented in the world [1], with devastating socio-economic, physical, and psychological consequences for communities. The pathophysiology of SARS-CoV-2 infection displays the predominance of hyperinflammation and immune dysregulation (cytokine release syndrome) in inducing multiorgan damage, predisposing patients to thrombotic and thromboembolic events due to endothelial cell activation and injury, platelet activation, and hypercoagulability [2]. Vaccines are the primary modality to prevent the disease from spreading. In 2020, an international race for developing vaccines against SARS-CoV-2 started [3]. However, like COVID-19, even the vaccines employed for its prevention have been associated with unexpected thrombotic events.

This paper aims to explore the complex relationships between COVID-19, its vaccines, and thrombotic diseases. Because of the low level of available evidence, and the continuous evolution of knowledge in this field, this is an interim document, based only on expert opinion consensus.

### 1.1. COVID-19 and Thrombosis

There are numerous relationships between cardiovascular disease and COVID-19. The presence of previous cardiovascular diseases is associated with a higher frequency of adverse outcomes in COVID-19, proportionally to the severity, extent, or symptoms of coronary lesions [4,5]. Conversely, among COVID-19 hospitalized patients, and in severe cases in general, a wide range of acute heart diseases, such as arrhythmias, fulminant myocarditis, acute heart failure, cardiogenic shock, pulmonary embolism (PE), or acute coronary syndromes (ACS), was commonly found [6,7,8,9,10,11,12,13,14]. Since March 2020, thrombo-embolic events have been increasingly described in the literature, with an incidence reaching 14% of hospitalized patients in surveillance wards and between 17% and 50% of patients in intensive care units [15]. Among 533 hospitalized patients with thrombotic events, an acute myocardial infarction (AMI) was present in more than half of the cases [16]. It is not surprising that COVID-19 can increase the ACS risk, as oxygen starvation, resulting from respiratory distress, and increased oxygen demands, occurring in response to infections, may cause a mismatch between oxygen supply and demands [17]. Local inflammation and hemodynamic changes may also increase the risk of the rupture of an atherosclerotic plaque [18,19]. An ST elevation MI (STEMI) may be the first clinical manifestation of COVID-19, but about a third of these patients do not present obstructive coronary artery disease [12,20,21] or angiographic signs of plaque rupture [20,22]. This finding highlights the potential role of endothelial dysfunction and hypercoagulation status [23,24,25]. Infection, hemodynamic stress of an acute critical pathology, inflammation (up to the typical hyper-reactive immune response that manifests itself with the cytokine storm) and fever, in fact, may favor a prothrombotic state, also interfering with the ability to dissolve thrombi, and may cause early or late instability and ruptures of coronary plaques and thrombosis [16,26,27]. High levels of IL-6, IL-1B, and IL-8, in fact, have been associated with plaque instability and increased thrombotic risk. Furthermore, IL-6 is involved in the stimulation of matrix-degrading enzymes such as matrix metalloproteinases, and may contribute to ACS development [28]. A STEMI could also be attributable to microthrombi formation [29].

Apart from ACS, other frequent COVID-19 thrombotic complications have been described in both arterial (stroke, acute limb ischemia, thrombosis of thoracic and abdominal aorta, mesenteric ischemia) and venous beds (deep venous thrombosis—DVT, cerebral venous sinus thrombosis—CSVT, and PE) [30,31,32,33,34,35,36]. PE represents the main complication, responsible for a five-fold increase in mortality [35]. In a meta-analysis of 102 studies involving 64,503 patients with COVID-19, the incidence of venous thromboembolism was 14.7% (95% CI 12.1–17.6%), while the rate of arterial thrombotic events was 3.9% (95% CI 2.0–3.0) [37]. According to other studies, the rate of venous thromboembolism varies from 25% to 69% [38]. Among hospitalized COVID-19 patients, a high incidence of alterations in inflammatory and coagulation biomarkers also correlates with a poor prognosis [39].

Studies to date suggest that the underlying pathophysiology of COVID-19-associated cardiac injury may be multi-factorial, as it can derive from both systemic perturbations (hyper-inflammation and thrombophilia) and potential direct cardiotoxic effects of SARS-CoV-2 due to disruption of the renin-angiotensin system, microangiopathy via endothelial cell/pericyte involvement (akin to parvovirus), or cardiomyocyte damage [40].

The rise in the incidence of thrombosis in large and small vessels can be explained by the presence of multiple factors, namely, the stasis of flow due to prolonged bed immobilization, vessel wall damage secondary to the loss of the normal thromboprotective state of the endothelium (due to inflammation and irritation caused by central venous catheters), hypercoagulable state caused by sepsis and endothelial activation due to the virus itself, thrombophilic inflammation responsible for the increase of von Willebrand factor and factor VIII, and neutrophil/platelet activation [41,42]. Venous thromboembolism is further favored by the hemodynamic effects of prolonged mechanical ventilation [38].

The activation of coagulation during systemic inflammation caused by different infectious agents is very complex and can occur through different mechanisms, involving polyphosphates derived from platelets activated by microorganisms, mast cells and factor XII, the complement system, and components of neutrophil extracellular traps (NETs), a mesh similar to a network that has the purpose of trapping viruses [43]. However, COVID-19-induced coagulopathy is different from that induced by sepsis, leading to extensive micro- and macro-vascular thrombosis and organ failure [44,45]. In addition, the type and rate of thrombosis can vary according to the cause of pneumonia: community-acquired pneumonia is more frequently complicated by arterial thrombosis, while an equal incidence of venous and arterial thrombosis occurs in SARS-CoV-2 [46]. 

It has been hypothesized that SARS-CoV-2 infection induces an immuno-thrombosis, in which neutrophils and activated monocytes interact with platelets and the coagulation cascade [38,47]. The main activation of the signal pathways to produce inflammatory cytokines are the toll-like receptors that recognize the presence of viral nucleic acids and the ACE-2 receptors, which the virus uses to infect cells. The coagulation alterations are mainly mediated by the activation of platelets [48]. The procoagulating effect of hypoxia should also be considered. A summary of hypothesized thrombotic mechanisms after COVID-19 is shown in Figure 1. It is possible to find a prothrombotic state also in long COVID-19, due to residual persistence of the blood chemistry of inflammation and procoagulative states [49].

The most frequently reported coagulation abnormality, especially in the most severe patients, is the elevation of D-dimer, but there are also increases in fibrinogen and its degradation products, PAI-1, and von Willebrand factor, as well as low levels of antithrombin III and antiphospholipid antibodies, known for their thrombophilic effect [29,39,45,50,51,52].

A simultaneous occurrence of CSVT and immune thrombocytopenic purpura has also been reported [53]. In COVID-19, thrombocytopenia is frequent and is associated with a worse prognosis. A meta-analysis showed that most severe cases of COVID-19 show a significant decrease in platelet counts (up to about 10,000) [46]. The pathogenesis of thrombocytopenia is probably related to the platelets’ overactivation by the complement through the generation of procoagulant microparticles and the insertion of C5b-9 in lytic quantities on platelets, in the absence of complement regulators [38]. Activated platelets also express a functionally active tissue factor (TF) that can trigger the coagulation cascade [54]. The resulting thrombosis leads to platelet consumption. 

The cascade of the uncontrolled multiplication of cytokines TNF-α, IL-1, and IL-6 present in patients with severe COVID-19, induced by complement and activated platelets, can also lead to a disseminated intravascular coagulation syndrome (DIC) [28,34,35,38]. In a meta-analysis of 14 studies, the incidence of DIC was 3% (95% CI: 1–5%, *p* < 0.001) and affected patients showed higher mortality (OR = 2.46, 95% CI: 0.94–3.99, *p* < 0.001) [55]. The pathophysiology of DIC associated with COVID-19 is very different from that associated with sepsis, including hemorrhages and macro- and micro-thrombosis [51]. In addition, in contrast to sepsis-induced DIC, where fibrinolytic activity is minimal, fibrinolysis is increased in COVID-19 [38]. 

### 1.2. Anti-COVID-19 Vaccines 

The natural history of COVID-19 can only be changed with the extensive use of vaccination. Favorable results from rigorous randomized, controlled phase III trials have been published for the Pfizer–BioNTech [56], Moderna [57], AstraZeneca/Oxford [58], and Johnson & Johnson/Janssen Cilag [59] vaccines, as well as the Russian Gam-COVID-Vac and the Novavax vaccine [60,61]; Table 1 shows the main COVID-19 vaccines.

The Pfizer–BioNTech and Moderna vaccines are based on messenger RNA (mRNA). AstraZeneca/Oxford uses a modified chimpanzee adenovirus to contain the gene for spike glycoprotein (S) production; Janssen Cilag uses the modified human serotype 26 adenovirus vector in a single administration, which encodes the complete S sequence by stimulating both neutralizing anti-S antibodies and other functional anti-S antibodies, as well as direct cellular immune responses. Sputnik uses two different adenoviruses for the two doses of vaccine, and Novavax has produced a protein-based vaccine containing tiny particles obtained from a recombinant version of protein S. Vaccines derived from chemically inactivated cultured viruses are produced by Sinopharm and Sinovac and are available in China [61].

To date, 240 candidate vaccines have been registered by the WHO, 63 of which are in the clinical evaluation phase, 177 in the preclinical phase, and 111 authorized for use in at least one country [61]. As of 26 December 2021, a total of 8,948,475,404 vaccine doses have been administered [1].

Medical practitioners are still required to make a special effort to promote the vaccination of patients with cardiovascular diseases. However, reports of thrombotic events in conjunction with some vaccines have caused much concern and even panic among the population and the medical community [61,62]. These serious, albeit rare, side effects related to vaccination, in our opinion, require further reflections, in anticipation of having to recommend vaccination even to patients returning from recent arterial and venous thrombotic episodes not related to COVID-19.

### 1.3. Anti-COVID-19 Vaccines and Thrombosis

In the initial phase III clinical trials [56,57,58,59], no major safety warnings, including thrombosis, were reported, apart from rare cases of anaphylaxis. In addition, a systematic review of the safety of vaccines in pivotal trials indicates that they are safe and without serious adverse events [63]. However, it is not surprising that new reports of adverse events emerge as long as more people are vaccinated and follow-ups become more extensive [64,65,66]. In March 2021, three descriptions of a new syndrome, characterized by thrombosis in unusual locations (CSVT, splenic vein thrombosis—SVT, thrombosis of the porta, mesenteric, or hepatic veins) and thrombocytopenia 4–28 days after the first dose of the AstraZeneca/Oxford vaccine were published [67,68,69] (11 patients in Germany and Austria, 23 in the United Kingdom, and 5 in Norway). These subjects were typically healthy or clinically stable, but about 40% of patients died, either from cerebral ischemia or overlapping hemorrhage. Most were women under the age of 50.

These reports were followed by many other articles on various events after administration of the AstraZeneca/Oxford vaccine, including DVT, PE or acute arterial thrombosis at various levels, cerebral arterial thromboembolism, and thrombotic microangiopathy [70,71,72,73,74,75,76,77,78,79,80,81,82,83,84,85,86,87,88,89,90,91,92,93,94,95,96,97,98,99,100,101,102,103,104,105,106,107,108,109,110,111,112,113,114,115,116,117,118,119,120,121,122].

Although adverse effects were observed more frequently for females younger than 60 years [123], the European medicine Agency (EMA) does not consider age and gender as significant risk factors, as the scarcity of data precludes robust estimates. Conversely, known risk factors are the use of estrogen-containing drugs and pregnancy [124].

Consequently, many regulatory agencies published the data of their surveillance systems on thrombotic events after COVID-19 vaccination [125,126,127,128,129,130,131,132,133,134,135,136,137,138,139] (Table 2).

After vaccination with the AstraZeneca/Oxford vaccine, 7 DIC cases were also observed in around 20 million subjects in the UK and Europe [140], and a link with the vaccine was considered possible. 

A second vaccine that has been associated with the appearance of thrombosis is the Janssen Cilag one [135,141,142,143,144,145,146], so it has been hypothesized that viral vectors could play a role. We do not have data on the thrombotic risk of another adenoviral vaccine, the Sputnik V [61].

Very rare cases of thrombosis have been related to mRNA vaccines [61,113,117,147,148,149,150,151,152,153,154,155,156,157,158,159,160]. In a study conducted at the Mayo Clinic on 68,266 subjects vaccinated with mRNA vaccines and as many controls, McMurry et al. showed that the incidence of TSVC is not higher in the vaccinated [161]. Finally, a very recent Israeli study on 884,828 individuals vaccinated with Pfizer–BioNTech showed only a high risk of myocarditis (RR 3.24; 95% CI 1.55–12.44), which was much lower than that related to COVID-19 (RR 18.28; 95% CI 3.95–25.12), but not of thrombotic events [162]. According to the US Centers for Disease Control and Prevention, myocarditis/pericarditis rates are ≈12.6 cases per million doses, mainly after the second dose of an mRNA vaccine, among individuals 12–39 years of age [163].

The lack of the total number of patients who have received a particular vaccine, and in particular, of reliable denominators stratified by age and sex, however, does not allow a direct comparison between the different vaccines [164]. In U.S., reporting rates for thrombosis with thrombocytopenia were 3.83 per million vaccine doses (Ad26.COV2.S) and 0.00855 per million vaccine doses (mRNA-based COVID-19 vaccines) [165]. Recent metanalyses suggest that approximately half of patients with thrombosis and thrombocytopenia syndrome present with CVST [166], vaccines against SARS-CoV-2 are not associated with an increased risk of thromboembolism, hemorrhage, and thromboembolism-/hemorrhage-related death [167], and the prevalence of thrombotic thrombocytopenia following ChAdOx1-S was 0.73 per 100,000 [168]. These data also suggest the importance of finding further mediators of this aberrant immune response beyond the adenoviral sequences or other components of the AstraZeneca and Janssen Cilag vaccines [139].

Regarding thrombocytopenia, about 60 cases, of which 2 were fatal, were reported in the UK after AstraZeneca/Oxford, and 34 cases (1 fatal) with BioNTech/Pfizer, while 195 cases were reported after BioNTech/Pfizer and Moderna vaccines in the U.S. Vaccine Adverse Event Reporting System (VAERS) [61,169]. Cases of thrombotic thrombocytopenic purpura, possible after each vaccine, have also been reported after BioNTech/Pfizer [170,171,172] and Janssen Cilag vaccines [173].

Regarding myocardial infarction, there are reports after AstraZeneca/Oxford [174] and, in a 96-year-old woman, after Moderna [175]. It is plausible that the stress of receiving the vaccine, as well as the reported adverse events (injection site pain, asthenia, nausea and vomiting, fever) may trigger an increased oxygen demand in the presence of an unknown coronary atherosclerotic burden. However, it is likely that a similar adult could have had a poor prognosis in case of infection with COVID-19. In a study of 126,661,070 vaccinated subjects, the incidence of heart attack increased with age (very rare in children, rare in women aged 30 to 54 years, uncommon in men and women aged 55 to 84 years, and common in those over eighty-five) [176].

Another point is the incidence of bleeding events. Out of more than 30 million vaccinated, the UK Medicines and Healthcare Products Regulatory Agency (MHRA) reported 267 hemorrhages (including 6 fatal) with AstraZeneca/Oxford, and 220 (9 fatal) with BioNTech/Pfizer, and in the VAERS database, out of more than 110 million vaccinated, 439 hemorrhagic events were reported with the BioNTech/Pfizer and Moderna vaccines [61].

### 1.4. Prognostic, Preventive, and Therapeutic Aspects

In a systematic review of the outcomes of patients with thromboembolic events following the AstraZeneca vaccine, 39 out of 146 patients died [177]. A recent systematic review and post hoc analysis, in which a total of 25 studies with 69 patients were included, investigated prognostic predictors in vaccine-associated thrombosis. Platelet nadir (*p* < 0.001), arterial or venous thrombi (χ^2^ = 41.911, *p =* 0.05), and chronic medical conditions (χ^2^ = 25.507, *p* = 0.041) were statistically associated with death. The ROC curve analysis yielded D-dimer (AUC = 0.646) and platelet nadir (AUC = 0.604) as excellent models for death prediction [178]. Additionally, in a multicenter British cohort study, CSVT is more severe in the context of thrombocytopenia [179]. In an international registry of consecutive patients with CVST within 28 days of SARS-CoV-2 vaccination from 81 hospitals in 19 countries [180], fibrinogen levels, age, platelet count, and the presence of intracranial hemorrhage (ICH) were significantly associated with mortality, and the FAPIC score comprising these risk factors could predict mortality [181]. In a prospective cohort study in the United Kingdom, the odds of death increased by a factor of 2.7 (95% CI 1.4 to 5.2) among patients with CSVT, by a factor of 1.7 (95% CI, 1.3 to 2.3) for every 50% decrease in the baseline platelet count, by a factor of 1.2 (95% CI, 1.0 to 1.3) for every increase of 10,000 units in the baseline D-dimer level, and by a factor of 1.7 (95% CI, 1.1 to 2.5) for every 50% decrease in the baseline fibrinogen level; the observed mortality was 73% among patients with platelet counts below 30,000 per cubic millimeter and ICH [182]. In a recent systematic review on thrombosis with thrombocytopenia after adenoviral vaccines, the mortality rate was 36.2%, and patients with suspected TTS, venous thrombosis, CVST, pulmonary embolism, or intraneural complications, patients not managed with non-heparin anticoagulants or i.v. immunoglobulins, receiving platelet transfusions, and requiring intensive care unit admission, mechanical ventilation, or neurosurgery were more likely to expire than recover [183].

Regarding prophylaxis with antithrombotic drugs, there is no scientific evidence to support the hypothesis that aspirin or low molecular weight heparin are effective in reducing the risk of thrombotic events in subjects undergoing vaccination against COVID-19 with adenoviral vaccines, in the face of a risk of serious adverse events, such as a greater, well quantifiable, and relevant hemorrhage [184]. It is obvious that these drugs may be continued in patients already treated.

Patients with thrombocytopenia after vaccination respond favorably to immunotherapy with intravenous steroids and immunoglobulins [169], whose possible benefits include blocking Fcγ receptor IIa (FcRγIIA), neutralizing anti-platelet factor-4 (PF4) antibodies by anti-idiotype antibodies, facilitating the catabolism of anti-PF4 antibodies, and modulating the immune cell compartment, including B cells that produce anti-PF4 [185].

### 1.5. Pathophysiological Hypotheses

The pathogenesis of hypercoagulability after vaccination remains poorly understood, and several questions are still open [186,187,188,189,190,191,192,193,194,195,196,197,198,199,200,201,202]. Both host and vaccine factors might be involved, with pathology, at least in part, being related to the vaccine-triggered autoimmune reaction [155].

Common findings in the cases described were the presence of elevated levels of D-dimer and antibodies against PF4 (identified by ELISA-based assays) and protein S, combined with thrombocytopenia but also hypofibrinogenemia, factor XIII deficiency, MTHFR C677T heterozygosis, and folate deficiency with increased levels of homocysteine and/or antiphospholipid antibodies [68,70,71,72,186,203,204,205,206]. 

Coagulopathies, including thromboses, thrombocytopenia, and other related side effects, are likely correlated to an interplay of the two components in the vaccine, i.e., the spike antigen and the adenoviral vector, with the innate and immune systems, which under certain circumstances can imitate the picture of a limited COVID-19 pathological picture [207,208]. Circulating platelets serve as a reservoir of immunomodulatory molecules [201]. However, no significant activation of fibrinogen-driven coagulation, plasma thrombin generation, or clinically meaningful platelet aggregation after AstraZeneca/Oxford or BNT162b2 vaccination was observed [209].

The mechanisms behind this are currently a subject of active research and include the following: (1) the production of PF4-polyanion autoantibodies; (2) adenoviral vector entry in megacaryocytes and the subsequent expression of spike proteins on platelet surfaces; (3) direct platelet and endothelial cell binding and activation by the adenoviral vector; (4) activation of endothelial and inflammatory cells by the PF4-polyanion autoantibodies; (5) the presence of an inflammatory co-signal; (6) the abundance of circulating soluble spike protein variants following vaccination [210].

The first important pathophysiological mechanism has been hypothesized by the Greifswald Working Group led by Andreas Greinacher. The constellation of thrombosis and thrombocytopenia has led to the hypothesis of a condition similar to heparin-induced thrombocytopenia (HIT), in which IgG-specific antibodies recognize the multimolecular complex between PF4-cationic and heparin polyanionic elements as foreign, causing multicellular activation and, in particular, the activation of monocytes and platelets through their FcRγIIA receptor, with a release of procoagulant metalloproteinases, as well as a direct activation of the endothelium by antibody complexes, leading to increased thrombogenicity with the release of selectin P and E, von Willebrand factor, interleukin 6, and thrombin, with consequent thrombocytopenia from platelet consumption and severe thrombogenicity [186,187,188,211]. The diagnosis of certainty of HIT requires the demonstration of the presence of anti-PF4-heparin antibodies. A similar mechanism could be implicated in the antiviral response to SARS-CoV-2, considering that antibodies to PF4-heparin have been detected in cases of patients with COVID-19. Terpos et al. [212], moreover, detected non-platelet activating anti-PF4 antibodies in 67% of the vaccinated individuals following the first dose of the AstraZeneca/Oxford vaccine. Vaccination can, therefore, probably induce the formation of antibodies against platelet antigens as part of the inflammatory reaction and immune stimulation; the adenoviral epitopes used in vaccines also have a strong affinity for PF4, mimicking the effect of heparin; this allows PF4 tetramers to cluster and form immune complexes through electrostatic interaction, which, in turn, causes FcγRIIa (also known as CD32a) massive dependent platelet activation [213], increased TF expression, and subsequent thrombin generation [190,191,204,214], regardless of the presence of heparin. However, it is unclear whether PF4 is a mere witness within an immune complex that activates platelets, or directly contributes to the formation of the thrombus. The delay in the production of these autoantibodies would explain the appearance of adverse reactions 4–14 days after vaccination [61]. Since none of the patients were exposed to heparin, the name autoimmune HIT was proposed for the syndrome [67]. Other proposed definitions are vaccine-induced prothrombotic immune thrombocytopenia (VIPIT) and vaccine-induced immune thrombotic thrombocytopenia (VITT), which is the most widely used. VIPIT, occurring after a vaccination, may pass into an asymptomatic state or it may have severe clinical complications as VITT [215]. Several international scientific societies have recently published recommendations on the diagnosis and management of VITT [64,216,217,218,219,220,221]. A suspect VITT should raise prompt testing for anti-PF4 antibodies [222].

Second, it has been hypothesized that even an accentuated immune response, a mechanism that mimics the effect of active COVID-19, could represent a thrombotic trigger [164]. The infection induces the activation of neutrophils and monocytes, with release of leukocyte DNA, which interact with platelets and the coagulation cascade leading to intravascular formation of thrombi in large and small vessels [192,193]. As mentioned before, during viral infections, and particularly in the case of SARS-CoV-2, one of the adaptive responses of the innate immune system (not selective but immediate) is the production by neutrophils of NET via IL-1β/NLRP3 inflammasome activation [223,224]. Although NET is useful and effective, numerous studies have shown its association with thrombosis [202]. The protein S, expressed by vaccines, also activates the complement system and can induce a cellular and humoral immune cascade against the virus favoring thrombosis [194]. Disproportionate inflammation can also increase endothelial adhesion and the release of TF, the true trigger of thrombin generation, a key enzyme in coagulation [187]. Although most of the reports on VITT have focused on the role of platelets, it is likely that VITT pathogenic antibodies bind and activate other cells that express FcγRIIa, notably, leucocytes and endothelial cells. The association between thrombocytopenia and often multiple thrombotic complications with a rapidly worsening clinical course is known to occur in other syndromes on autoimmune basis, such as antiphospholipid syndrome, already demonstrated in COVID-19 [196,206], or thrombotic thrombocytopenic purpura.

According to a further hypothesis [191,225], an accidental injection of the vaccine into a vein, even in small quantities, or multiple exudations over time, can culminate in high levels of adenoviruses in the blood, which, although not replicating, can infect permissive cells such as epithelial or endothelial ones and fibroblasts, which can process large amounts of S glycoproteins, leading to high levels of antigens against them. This is not possible in the case of mRNA vaccines since lipid nanoparticles cannot survive in the enzymatically hostile plasma environment and are rapidly eliminated from the reticulo-endothelial system. 

Genetic vaccines could instead directly infect platelets and megakaryocytes, causing mRNA translation and intracellular synthesis of S proteins, which would cause an autoimmune response against these elements, causing reticulo-endothelial phagocytosis and direct lysis of CD8 T cells. When a vaccinated cell dies or is destroyed by the immune system, in addition, debris can release a large amount of whole or fragmented S proteins into the blood. In a subject with previous SARS-CoV-2 infection or with cross-reactive antibodies to common coronaviruses, a large volume of immune complexes can form shortly after vaccination with adenovirus-based vaccines, but also with mRNA [190]. IgG against these immune complexes can be glycosylated in an aberrant way (e.g., afucoselate) as is the case in the most severe cases of COVID-19.

Finally, it is known that SARS-CoV-2 uses ACE-2 as a Trojan horse to invade target cells. Vaccines have the potential to interact with ACE-2, promoting its internalization and degradation, a phenomenon also observed in platelets, in which subunit 1 of protein S, but not subunit 2, binds to ACE-2, inducing a dose-dependent facilitation of aggregation and release of adenosine triphosphate [192]. The loss of ACE-2 receptor activity from the outer side of the cell membrane, mediated by the interaction between ACE-2 and spike proteins, results in less angiotensin inactivation, which increases thrombotic risk [189,194].

A distinctive feature of the SARS-CoV-2 spike protein is its ability to efficiently fuse cells, thus producing syncytia found in COVID-19 patients; this ability may enable spike to cause COVID-19 complications as well as side effects of COVID-19 vaccines [226].

Finally, in a recent study, both adenoviral and mRNA vaccines enhanced inflammation and platelet activation, though adenoviral vaccination induced a more pronounced increase in several inflammatory and platelet activation markers compared to mRNA vaccination, and post-vaccination thrombin generation was higher following adenoviral vaccination compared to mRNA vaccination [227]. Additionally, no difference in either the PF4 antibody level or the proportion of individuals with positive PF4 antibodies was observed between the vaccine groups [227].

All, or many, of these conditions should be present to trigger platelet and thrombosis, which explains the rarity of these cases [199,200]. The different pathophysiological hypotheses are illustrated in Figure 2. 

It is not clear why this immunogenic thrombosis occurs in the cerebral or splanchnic vessels, that is, whether it is correlated with the localization of the antigen or with the vascular response. The presence of specific polyanionic antigens in the mentioned vascular sites could be a possible explanation. Venous drainage of microbiotic-rich areas in the nose and intestines, which can trigger local endovascular immunity with engagement of autoantibodies directed towards PF4-microbiote complexes, could play an additional role [228]. 

Finally, a myocardial infarction could be also provoked by vaccine-induced allergic vasospasm, as in Kounis syndrome [229,230]. A mRNA COVID-19 vaccine-related anaphylactoid reaction and coronary thrombosis has been described [231].

### 1.6. Regulatory Aspects 

On 15 March 2021, due to the cited reports of thrombosis and some suspicious deaths, several European health institutions suspended the use of the AstraZeneca/Oxford vaccine throughout the national territory [64,232,233]. On 18 March, in the UK, the MHRA stated that the evidence does not suggest that thrombosis is caused by the Astrazeneca/Oxford vaccine, while the EMA concluded that a causal link with the vaccine was possible [35]. The EMA has also decided to include information on thrombotic risk in package leaflets, warning patients and doctors to be vigilant about the potential appearance of symptoms. The EMA Pharmacovigilance Risk Assessment Committee (PRAC), after the necessary checks, readmitted the use of the AstraZeneca/Oxford vaccine on 19 March [125,232]. On 27 March, the EMA, the COVID-19 subcommittee of the WHO Global Advisory Committee on Vaccine Safety (GVACS), and the MHRA reviewed the risk of thrombosis after vaccination with AstraZeneca/Oxford, again agreeing that the benefits outweigh the risks. On 7 April 2021, including information from the EMA and MHRA (which advised to offer young people under 30 an alternative to AstraZeneca/Oxford if available), the GACVS published an interim statement that the causal relationship between the vaccine and the onset of thrombi and platelet disease is plausible but not confirmed, and in the UK, an age restriction regarding the AstraZeneca/Oxford vaccine for people under 30 years of age was introduced. On the same date, the Italian Ministry of Health recommended the preferential use of the AstraZeneca/Oxford vaccine in people over 60 years of age, considering the low risk of thromboembolic adverse reactions, in the face of the high mortality from COVID-19 in an age group in which the vaccine is certainly effective in reducing the risk of serious disease, hospitalization, and death related to COVID-19 [132]. In addition, it has been stated that it is not possible to make recommendations regarding the identification of specific risk factors, and that preventive treatments of the aforementioned thrombotic episodes are not identifiable. On 7 May, in the UK, a restriction of AstraZeneca/Oxford vaccine was introduced for people under 40 years of age. On 22 July 2021, the Advisory Committee on Immunization Practices reviewed updated benefit-risk analyses after Janssen Cilag and mRNA COVID-19 vaccination, and concluded that the benefits outweigh the risks for rare, serious adverse events after COVID-19 vaccination [234]. Finally, in December 2021, the U.S. Advisory Committee on Immunization Practices voted unanimously (15 to zero) for a recommendation for the preferential use of mRNA COVID-19 vaccines over the Janssen COVID-19 vaccine for the prevention of COVID-19 for all persons aged ≥18 years [235].

### 1.7. Final Considerations

Vaccines against SARS-CoV-2 have been used for a short time, and knowledge about their clinical manifestations is constantly evolving [34,236]. Data on their long-term effects, interactions with other vaccines, use in immunocompromised subjects, and those with comorbidities (e.g., hematological, autoimmune or inflammatory disorders) are lacking [237]. Therefore, careful surveillance and long-term follow-up studies are needed [136]. In this regard, it is important to remind all cardiologists, but also hematologists, and in some cases, internists and vascular surgeons, to report all suspected adverse reactions associated with the use of COVID-19 vaccines, in accordance with the respective national reporting system. 

Unfortunately, part of the population still hesitates to recognize the dangers associated with SARS-CoV-2, comparing them to past influenza epidemics, ignoring the fact that mortality continues to rise in the world despite strict hygiene and lock-down measures. The phenomenon of denial is important, and has been influenced by reports of side effects, especially thrombotic. The vaccine hesitancy is a complex phenomenon that is driven by individuals’ perceptions of safety and the efficiency of the vaccines [238].

It should be remembered that the incidence of severe thrombotic events appears low (1/100,000–1/1,000,000 vaccinated subjects) [130,161,162,204,239,240]. In a real-world evidence-based study, which retrospectively analyzed a cohort of 771,805 vaccination events across 266,094 patients in the Mayo Clinic Health System between 2017 and 2021, CVST was rare and not significantly associated with COVID-19 vaccination [241]. However, clinical trials that tested the effectiveness of vaccines included only SARS-CoV-2-negative subjects. The possibility cannot be excluded that the vaccination of an increasing number of subjects may cause an unexpected thrombotic and inflammatory reaction in subjects predisposed by a previous infection [189].

Pharmacovigilance reports, however, contain administrative and uncontrolled data that are undoubtedly useful, but they cannot and should not support hypotheses about causal relationships. Early signs of rare side effects during pharmacovigilance that can lead to severe outcomes should not be set aside solely based on statistical prevalence but require extensive scientific studies and a correlation with the clinic to rule out a potential causal link [196].

When evaluating for the purpose of making decisions about the use of drugs, it is important to consider the natural history of pathologies, based on pre-pandemic incidence rates in the general population. In the general population, regardless of each vaccination, the annual incidence of venous thrombosis and cerebral thrombosis are, respectively, 1.2/1000 and 1.2/100,000 [61]. In the Danish National Patient Registry [242], cases of venous thromboembolism (DVT, PE, hepatic vein thrombosis, mesenteric, portal vein thrombosis, renal or hollow vein thromboembolism, *migrans thrombophlebitis*, intracranial vein thrombosis) were identified in all adults between 2010 and 2018, calculating an incidence of venous thromboembolism of 1.76 per 1000 patients per year (0.95/1000 between 18 and 64 years). In the 5 million Danish inhabitants (corresponding to the number of subjects who received AstraZeneca/Oxford vaccines in Europe as of 10 March) the incidence of venous thromboembolism corresponds to 169 cases per week in all adults (91 from 18 to 64 years). In contrast, only 30 cases of thromboembolic events have been reported after AstraZeneca/Oxford vaccination, which, therefore, does not appear to increase the incidence rate of venous thromboembolism compared to the natural one. However, these data cannot rule out the possibility that some venous thrombotic events after AstraZeneca/Oxford are caused by the vaccine, as they occurred after a short interval of time.

The risks and benefits of current vaccines must be compared with the real possibility of contracting the disease and developing long-term complications and sequelae based on the available clinical evidence and avoiding unjustified bias [243,244]. 

In fact, thromboembolic complications of COVID-19 are much more frequent (6 to 28% of cases) [131,232,239,245,246,247]. In a study by Taquet et al. [247], the incidence of CSVT in the two weeks after a COVID-19 diagnosis (42.8 per million people, 95% CI 28.564.2) was significantly higher than in a matched cohort of people who received an mRNA vaccine (RR = 6.33, 95% CI 1.8721.40, *p* = 0.00014) or in patients with influenza (RR = 2.67, 95% CI 1.046.81, *p* = 0.031), and the incidence of peripheral thrombosis after COVID-19 diagnosis (392.3 per million people, 95% CI 342.8448.9) was significantly higher than in a matched cohort of people who received an mRNA vaccine (RR = 4.46, 95% CI 3.126.37, *p* < 0.0001) and in patients with influenza (RR = 1.43, 95% CI 1.101.88, *p* = 0.0094). Comparing data from the US Centers for Disease Control and Prevention, the Nationwide Inpatient Sample, and the Society of Vascular and Interventional Neurology COVID-19 registry, Bikdeli et al. [239] highlighted that the incidence of CSVT was 0.9/million in vaccinated people, 2.4/million in the general population and 207.1/million in COVID-19 patients.

Risks versus benefits varied significantly between age groups and transmission levels. Across different scenarios, benefits of adenoviral vaccination in people 55 years and older exceeded the risk of death from COVID-19. In young adults, the risks were at least of a similar magnitude as the benefits [248]. For example, for every million doses of the Janssen Cilag vaccine administered to women aged 18 to 48 years, 297 hospital admissions, 56 admissions to intensive care, and 6 deaths related to COVID-19 are avoided, compared to 7 cases of thrombosis [246]. Under a high transmission rate, deaths prevented by AstraZeneca/Oxford vaccine far exceed deaths from VITT (by 8 to > 4500 times depending on age). The probability of dying from COVID-19-related atypical severe blood clots was 58–126 times higher (depending on age and sex) than dying from VITT [249]. Excess deaths due to the interruption of the AstraZeneca vaccination campaign in France and Italy largely overrun those due to thrombosis, even in worst case scenarios of frequency and gravity of the vaccine side effects [250]. For AstraZeneca/Oxford vaccination itself, a recent Italian study showed that the benefits outweigh the risks as early as the age of 30 [251].

Authorities, media, and the population should also be reminded that thrombotic risks are accepted in the modern lifestyle; millions of women use contraceptives that increase their thrombotic risk by 3 to 5 times, and the absolute risk of having a venous thrombosis after an air trip lasting more than 4 h is 1/4600, much higher (50–100 times) than that of having a CSVT after vaccination [61,252]. Clearly, in VITT, mortality differs from classic DVT, reaching 40% [61].

The ESC Patient Forum published information on the COVID-19 vaccine for heart patients on 12 April, reiterating the importance that everyone can receive it due to the high risk of complications [253]. Trials with COVID-19 vaccines have included patients with heart disease without demonstrating serious effects; no vaccine and cardiological drug interactions are reported and there is no evidence to suggest that they are contraindicated in heart disease. Furthermore, due to the immunogenic nature of thrombosis, patients with a history of thrombosis and/or known thrombophilia do not have an increased risk after AstraZeneca vaccination [64]. In fact, it is estimated that about 5000–6000 subjects per 100,000 vaccinated are carriers of these coagulative abnormalities [254], which clearly contrasts with the extreme rarity of the most serious thrombotic complications observed. COVID-19 vaccines seem safe for patients with previous CVST [255]. There is also no evidence that thrombosis at typical sites (lower limbs, pulmonary embolism) is more common after adenoviral vaccines than in the general population stratified by age. Therefore, there are no elements to contraindicate vaccination in patients returning from a recent thrombotic event or, in particular, from an AMI, but is opportune, giving preference to mRNA vaccines in younger classes, particularly in women, in accordance with the indications of the regulatory authorities.

There are no reliable data on the risk related to the booster dose. According to the hypothesis of a hyperactivity of coagulation induced by the vaccine, it is reasonable to expect that, with the first administrations, there has already been the so-called “depletion of susceptible”, or a sort of selection of subjects who, for unknown reasons, are more exposed to the action of these hypothetical prothrombotic mechanisms, and that, therefore, any adverse manifestations are even rarer following the second dose [256]. According to the hypothesis of the production of autoantibodies, re-exposure to the vaccine could instead lead to important clinical manifestations in some subjects who, at the first dose, had already activated an abnormal immune response, even if it was clinically not evident [67]. As of 12 May, 15 cases of atypical thrombosis with thrombocytopenia have been reported by the English MHRA for about 9 million second doses of AstraZeneca/Oxford administered, which would seem to correspond to a weaker signal than that found for the first doses and, in any case, is definable as very rare, supporting the “depletion of susceptible” hypothesis.

## 2. Conclusions

The rapid availability of vaccines, effective in limiting transmission and severe forms of the disease, has emerged as the only solution for controlling the SARS-CoV-2 pandemic. Careful surveillance and long-term follow-up studies on vaccines are needed. Unfortunately, part of the population still hesitates to recognize the dangers associated with SARS-CoV-2. Healthcare professionals remain the most appropriate advisers regarding vaccination decisions and must be supported to provide reliable and credible information. Table 3 suggests, for example, what to avoid in case of vaccination [61]. It should be remembered that the incidence of severe thrombotic events appears low. The risks and benefits of current vaccines must be compared with the real possibility of contracting the disease and developing long-term complications. Authorities, media, and the population should also be reminded that thrombotic risks are accepted in the modern lifestyle. All scientific societies emphasize the value of continuing vaccination programs to protect patients from severe forms of COVID-19 and to slow the circulation of the virus and its variants [257]. Vaccine hesitancy risks regressing progress in infectious disease control. Abstention is not an option, as it results in a failure to assist a large population that remains in danger. Action, with increased vigilance, is the best solution in our public health mission [61].

## Figures and Tables

**Figure 1 jcm-11-00948-f001:**
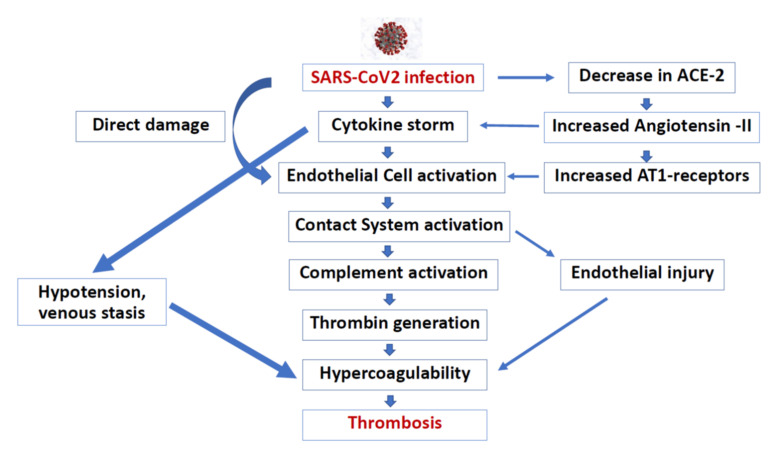
Hypothesized thrombotic mechanisms in COVID-19.

**Figure 2 jcm-11-00948-f002:**
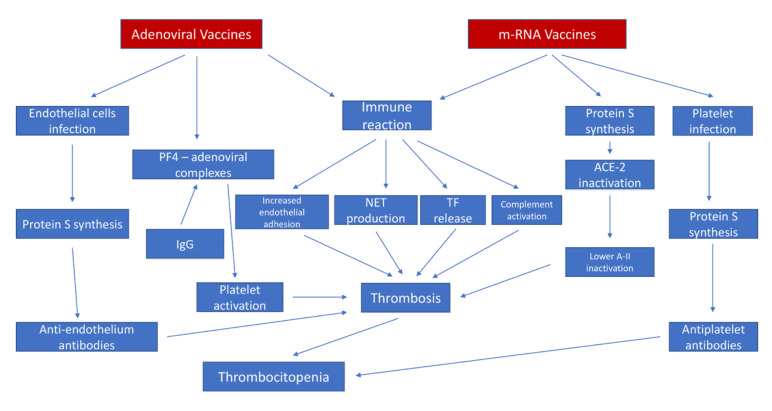
Hypothesized thrombotic mechanisms after COVID-19 vaccination.

**Table 1 jcm-11-00948-t001:** Main anti-COVID-19 vaccines.

Name	Company	Types
BNT162b2 (Comirnaty^®^)	Pfizer (New York, NY, USA)—BioNTech (Mainz, Germany)	mRNA
mRNA-1273 (Spikevax)	Moderna (Cambridge, MA, USA)	mRNA
NVX-CoV2373 (Nuvaxovid)	Novavax (Gaithersburg, MD, USA)	Recombinant nanoparticles
AZDI222 (Vaxzevria^®^, Covishield)	AstraZeneca (Oxford, UK)	Adenovirus vector ChAdOx1
Ad26.CoV2.S	Janssen Biotech Cilag, Johnson & Johnson (Raritan, NJ, USA)	Adenovirus vector Ad26.CoV2.S
Gam-COVID-Vac (Sputnik V)	Gamaleya Institute (Moscow, Russia)	Adenovirus vector Ad26 and Ad5 CoV2-S
ConvideciaTM	CanSino Bio (Tianjin, China)	Adenovirus vector Ad5-nCoV
CoronaVac	Sinovac Biotech (Beijing, China)	Inactivated virus
BBIBP-CorV	Beijing Institute of Biological Products (Beijing, China)	Inactivated virus

**Table 2 jcm-11-00948-t002:** Thrombotic events after COVID-19 vaccination in main surveillance systems.

Source	Vaccine	Location	Surveillance System and Results Description
Krzywicka K et al. [126]	ChAdOx1BNT162b2 mRNA-1273	Europe	EudraVigilance database (EMA). Until 8 April 2021, 213 CVST cases were identified: 187 after ChAdOx1 and 26 after a mRNA vaccination (25 with BNT162b2, and one with mRNA-1273). Thrombocytopenia was reported in 107/187 CVST cases (57%, 95% CI 50–64%) in the ChAdOx1 group, in none in the mRNA vaccine group (0%, 95% CI 0–13%), and in 7/100 (7%, 95% CI 3–14%) in a pre-COVID-19 group with CVST.
Cari L et al. [127]	ChAdOx1 BNT162b2	Europe	EudraVigilance database (EMA). Frequency of SAEs up to 16 April 2021, related to thrombocytopenia, bleeding, and blood clots. ChAdOx1 administration was associated with a much-higher frequency of SAEs (33 and 151 SAEs/1 million doses in BNT162b2 and ChAdOx1 recipients, respectively). When considering SAEs related to cerebral/splanchnic venous thrombosis, and/or thrombocytopenia, 4 and 30 SAEs and 0.4 and 4.8 deaths/1 million doses were observed for BNT162b2 and ChAdOx1 recipients, respectively.
Van de Munckhof A et al. [134]	All	Europe	EudraVigilance database (EMA). Until 13 June 2021, 270 cases of CVST with thrombocytopenia were identified, of which 266 (99%) occurred after adenoviral vector SARS-CoV-2 vaccination (ChAdOx1, *n* = 243; Ad26.COV2.S, *n* = 23).
Cari L et al. [128]	ChAdOx1Ad26.COV2.S BNT162b2	Europe	EudraVigilance database (EMA). Severe adverse events (SAEs) documented in the young-adult (18–64 years old) and older (≥65 years old) vaccine recipients until 23 June 2021. Comparison between the frequency of SAEs and SAE-related deaths in adenoviral vs. BNT162b2 vaccine recipients demonstrated: (1) adenoviral vaccine recipients had higher frequencies of not only SAEs caused by venous blood clots and hemorrhage, but also thromboembolic disease and arterial events, including myocardial infarction and stroke; (2) a correspondingly higher frequency of SAE-related deaths in both young adults and older adults. Comparison between the frequency of SAEs demonstrated a lower frequency of thrombocytopenia and SAEs in young adults and higher frequency in older Ad26.COV2 recipients.
Abbattista M et al. [130]	All	Europe	EudraVigilance database (EMA). Data between 1 January and 30 July 2021. TSVC rate, for one million vaccinated, of 1.92 for BNT162b2, 5.63 for mRNA-1273, 21.6 for ChAdOx1, and 11.48 for Ad26.COV2.S.
www.aifa.gov.it (accessed on 31 October 2021) [132]	All	Italy	National Pharmacovigilance Network. Reports of SAEs following vaccination from 27/12/2020 as of 26/09/2021. A total of 101,110 SAEs in a total of 84,010,605 doses of vaccine. The number of cases of cerebral or atypical venous thrombosis with thrombocytopenia is very low, with results of less than 1 case per 1,000,000 doses administered.
Gras-Champel V et al. [133]	ChAdOx1	France	French Network of Regional Pharmacovigilance Centers. A total of 27 cases of severe thrombosis (24 CVST, 2 SVT, and 1 EP with DIC) in 3,300,000 subjects.
Hippisley-Cox J et al. [131]	ChAdOx1 BNT162b2	UK	Office for National Statistics and hospital admission data from the United Kingdom’s health service. Data were obtained for approximately 30 million people vaccinated in England between 1 December 2020 and 24 April 2021. Increased risk of thrombocytopenia after ChAdOx1 vaccination (incidence rate ratio 1.33, 95% CI 1.19 to 1.47 at 8–14 days); increased risk of venous thromboembolism after ChAdOx1 vaccination (1.10, 1.02 to 1.18 at 8–14 days); increased risk of arterial thromboembolism after BNT162b2 vaccination (1.06, 1.01 to 1.10 at 15–21 days). The risks of most of these events were substantially higher and more prolonged after SARS-CoV-2 infection than after vaccination in the same population.
Andrews NJ et al. [134]	ChAdOx1 BNT162b2	UK	Hospital admissions for cerebral venous thrombosis, other venous thrombosis or thrombocytopenia between 30 November 2020 and 18 April 2021 were linked to the national COVID-19 immunization register, showing an increased risk of thrombotic episodes and thrombocytopenia in adults under 65 years of age within a month of a first dose of ChAdOx1 vaccine but not after the vaccine.
See I et al. [135]	Ad26.COV2.S	USA	Vaccine Adverse Event Reporting System (VAERS) from 2 March to 21 April. With 8 million doses practiced, 15 reports of thrombosis, located in various unusual venous sites and arterial ones, all in blank women from 18 to 60 years, of which 7 had at least 1 risk factor (obesity, hypothyroidism, contraceptives)
Shay DL et al. [136]	Ad26.COV2.S	USA	Morbidity and Mortality Weekly Report, as of 30 April 2021. A total of 17 cases of thrombosis in atypical sites associated with thrombocytopenia in 7.98 million doses administered
www.fda.gov (accessed on 31 October 2021) [137]	BNT162b2 mRNA-1273	USA	VAERS database. A total of 161 classical thrombosis events in 125 million subjects vaccinated with mRNA-1273 and 153 million vaccinated with BNT162b2.
Klein NP et al. [138]	BNT162b2 mRNA-1273	USA	Vaccine Safety Datalink. The 10,162,227 vaccine-eligible members of 8 participating US health plans were monitored from 14 December 2020, through 26 June 2021. A total of 11,845,128 doses of mRNA vaccines (57% BNT162b2) were administered to 6.2 million individuals. The incidence of events vs. controls per 1,000,000 person-years was 1612 vs. 1781 (RR, 0.97; 95% CI, 0.87–1.08) for ischemic stroke and 935 vs. 1030 (RR, 1.02; 95% CI, 0.89–1.18) for AMI.
Smadja DM et al. [139]	ChAdOx1BNT162b2 mRNA-1273	World	Global Database for Individual Case Safety Reports (VigiBase) of Uppsala (Sweden), the largest pharmacovigilance register in the world, between 13 December 2020, and 16 March 2021: In out of 361,734,967 subjects who had received the three vaccines (population 15 times greater than that reviewed by EMA), 2161 thrombotic events were reported, with a rate of 0.21 (95% CI: 0.19–0.22) cases/million vaccinated subjects; of these, 795 were venous and 1374 arterial, with a rate of 0.075 (95% CI: 0.07–0.08) and 0.13 (95% CI: 0.12–0.14), respectively). A total of 1197 cases occurred after the BNT162b2, 325 after mRNA-1273, and 639 after ChAdOx1 vaccines. Venous events were lower than arterial events for mRNA vaccines, while the opposite was true for ChAdOx1 (52.2% vs. 48.2%).

**Table 3 jcm-11-00948-t003:** What to avoid in case of COVID-19 vaccination (from [61], modified).

Systematic premedication with low molecular weight heparin, direct oral anticoagulants, or aspirin
Systematic screening for thrombophilia
Systematic evaluation of PF4 antibodies after the vaccine
Systematic monitoring of changes in D-dimer
Systematic use of venous echo-doppler examinations after the vaccine

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
