# Peer review of "COVID-19, Vaccines, and Thrombotic Events: A Narrative Review"

_jcm, 2022, doi:10.3390/jcm11040948_

Round 1

Reviewer 1 Report

The authors presented information about the connection between COVID-19, vaccines and thrombosis. The article is good and I suggest accepting it after some corrections:

  1. In the Introduction, should be more information about vaccines, both used in patients and all time being in trials. Please cite here some articles about types of vaccines, eg. https://pubmed.ncbi.nlm.nih.gov/33408775/
  2. The authors should add a second figure, in which will be presented thrombotic mechanisms in patients with COVID-19, and long-COVID.
  3. For me, in the article is lack of Conclusions, it is 2-4 short sentences, concluding the whole article. Authors presented "1.7. Final considerations", but it has 3 pages and is not conclusion.

Reviewer 2 Report

In this narrative review, Dr. Abrignani and colleagues discussed the thrombotic complication of COVID-19 and the subsequent thrombosis risk following COVID-19 vaccination. Overall, this is a comprehensive and very thoroughly discussed manuscript. Some data contains speculation but it is understandable considering little what we know about this disease today. Nonetheless, I have some comments and suggestions to incorporate:

  • In the Introduction, the authors need to spend a few sentences to give introduction on COVID-19 pathophysiology in general. There has been many reviews about it so just provide the highlights, for example the presence of cytokine release syndrome, as discussed for example in this manuscript (https://pubmed.ncbi.nlm.nih.gov/34321903/).
  • In the section "Anti-COVID-19 vaccines and thrombosis", please add these data if haven't been done. 
    • Palaiodimou et al. https://pubmed.ncbi.nlm.nih.gov/34610990/
    • Matar et al. https://pubmed.ncbi.nlm.nih.gov/34980833/

    • Uaprasert et al.  https://thrombosisjournal.biomedcentral.com/articles/10.1186/s12959-021-00340-4

    •  

      Chan et al.  https://www.medrxiv.org/content/10.1101/2021.05.04.21256613v1 

  • Since this is a rapid-progressing topic, please double check if there are new data out there that are relevant to be included. 
  • Line 28: "...pandemic that has affected millions of people worldwide, is associated with cardiovascular complications, including such as both venous and..."
  • Line 29: "... may promote the release ..."
  • "This point-of-view aims at providing a comprehensive review..." I think "narrative review" is more appropriate. 
  • "helping cardiologists to offer an up-to-date and evidence-based counseling to their often-alarmed patients with acute or chronic cardiovascular thrombotic events" I think the authors should aim to capture the attention of broader specialties, as thrombosis is not only managed by cardiologists, but also hematologists and in some cases, internists. Also, vascular surgeons would be involved if there is acute limb ischemia requiring revascularization. So, please change the target readers from cardiologists to medical practitioners. 
  • Similarly, consider replacing "cardiologists" in line 165 etc. 
  • The authors need more keywords. Please add some more. The max is 10 so use it to ensure that this manuscript is discoverable in search engine. 
  • "Since the beginning of the SARS-CoV-2 pandemic and consequent (Coronavirus Disease 2019, COVID-19),..." I don't think this is correct. Please rephrase. 
  • "more than 279 million cases and over 5,3 million deaths" please specify the date of data retrieval. 
  • Line 49: "...consequences for the community. COVID-19 and the adenovirus-based..." Between these two sentences, please add a sentence telling that vaccine is the primary modality to prevent disease spreading. 
  • Line 52: "Because of the low level of evidence available, and..."
  • The title of Table 3 could be changed into "What to avoid not to do in case of COVID-19 vaccination" to avoid confusion or misreading. 
  • In general, considering the extent of this review, some typos and grammatical errors could still be present. Please double-check everything. 

Round 2

Reviewer 2 Report

Thanks for the responses. I only have one minor suggestion but this can be addressed later. In Line 51, "to now" can be changed to "to date". 

This manuscript is a resubmission of an earlier submission. The following is a list of the peer review reports and author responses from that submission.